# New High-Throughput Method for Aluminum Content Determination in Vaccine Formulations

**DOI:** 10.3390/vaccines13020105

**Published:** 2025-01-22

**Authors:** Lorenzo Di Meola, Daniela Pasqui, Chiara Tigli, Stephen Luckham, Silvio Colomba, Marilena Paludi, Maxime Denis, Angelo Palmese, Daniela Stranges, Agnese Marcelli, Alessio Moriconi, Malte Meppen, Carlo Pergola

**Affiliations:** GSK, Via Fiorentina 1, 53100 Siena, Italy

**Keywords:** vaccines, MenB, adjuvant, aluminum, Neisseria meningitidis

## Abstract

Objective: This manuscript describes an innovative, non-destructive, high-throughput method for the quantification of aluminum hydroxide in aluminum-adjuvanted vaccines, eliminating the need of reagents and providing real-time results. The method is based on a spectrophotometric principle, and several model proteins were studied and tested with the aim to simulate the behavior of aluminum-adjuvanted antigens. Methods: As a proof of concept, the MenB vaccine was used, and the titration of aluminum hydroxide (AH) with ethylenediaminetetraacetic acid (EDTA) was used as an orthogonal reference, as it is one of the current release methods for the content determination of aluminum-hydroxide-adjuvanted vaccine drug products (DPs). The factors influencing the spectrophotometric analysis, such as different plate 96/well containers, variation in the sedimentation of the suspension due to component addition errors during formulation, and batch-to-batch variation were studied to assess the method’s robustness. Five concentration levels (ranging from 2.0 to 4.0 mg/mL AH) with two different batches of aluminum hydroxide were each measured with independent preparations performed by three different operators, for a total of four sessions/operator and 20 formulations/session. An in-depth statistical study was carried out with generated data to assess the precision (in terms of intermediate precision and repeatability), accuracy, linearity, and specificity of the method. Results: The novel spectrophotometric method and the official release one (potentiometric) yielded comparable results, demonstrating the potential of this new method as a release test for AH-adjuvanted products. A simple calibration curve enabled the measurement of samples in a 96-well plate in just a few minutes. Conclusions: We developed a novel method for Aluminum concentration determination in Aluminum-containing pharmaceutical products, like alum-adjuvanted vaccines. This method is fast, completely automatable, and as precise and accurate as already-in-place release methods.

## 1. Introduction

Aluminum-based adjuvants (alums) including aluminum hydroxide (AH) and aluminum phosphate (AP) are the most commonly used adjuvants in current commercial vaccines [1]. These adjuvants are used in vaccines to both stimulate the immune response and facilitate a depot effect. In order to ensure the expected quality of the final DP, regulatory authorities require the evaluation of alum content in the final vaccine products. The maximum amount of aluminum content allowed in a vaccine for human use is 1.25 mg per dose in Europe and 0.85 mg per dose in the USA [2]. Aluminum adjuvants are used in various vaccines, including those for hepatitis A, hepatitis B, diphtheria-tetanus-pertussis (DTaP), Haemophilus influenzae type b (Hib), and pneumococcal disease. Vaccine manufacturers therefore routinely perform quality control analyses to determine aluminum content, typically via potentiometric titration [3]. In this work a simple, non-destructive, high-throughput method was developed to determine the AH content by spectrophotometric analysis for supporting both product/process development studies and potential application in the final release testing of aluminum-adjuvanted vaccines. The method can determine the AH content when used in complex vaccinal formulations containing different antigens. We focused on the quantification of aluminum in the product Bexsero (Bexsero is a trademark owned by or licensed to the GSK group of companies), a multicomponent protein-based vaccine against meningococcal group B (MenB) [4,5,6,7], The active components of the vaccine are the recombinant proteins Neisseria heparin-binding antigen, a factor H-binding protein, and Neisseria adhesin A, and the outer membrane vesicles, OMVs, expressing porin A and porin B, produced by fermentation of the Neisseria meningitidis strain NZ98/254 [5,8,9,10]. All active components of Bexsero are adsorbed on AH, and the aluminum content is a critical quality attribute of the product and is analyzed as part of the DP release testing panel. The conventional methods for determining aluminum content, such as inductively coupled plasma optical emission spectrometry (ICP-OES) and atomic absorption spectroscopy (AAS), are known for their precision and accuracy but have significant limitations [11]. They are time-consuming, labor-intensive, require specialized equipment, and necessitate trained personnel. For example, ICP-OES is acclaimed for detecting trace aluminum ions in complex biological matrices but demands meticulous sample preparation, precise calibration, and technical expertise. Similarly, AAS provides precise measurements by analyzing the light absorption of free aluminum ions yet requires extensive preparation and is not practical for rapid, large-scale analyses. Both methods, while effective, lack the speed and efficiency required for the crucial stages of vaccine development and process optimization. In the vaccine development phase, especially when formulating new candidates or optimizing production processes, the rapid analysis of aluminum content is essential to refine formulations and ensure compliance with safety and efficacy standards. This becomes particularly critical during public health emergencies, such as pandemics, where time constraints necessitate quick adjustments. Traditional methods, which can take hours or days for results, do not align with the high-throughput demands of modern vaccine development, where swift testing and the adjustment of formulations are necessary. In response to these challenges, we have developed an innovative spectrophotometric method for quantifying aluminum content in vaccines. This new approach, characterized by its speed, simplicity, and practicality, offers a significant advancement over traditional techniques. We tested several model proteins to simulate the behavior of aluminum-adjuvanted antigens in the MenB vaccine, aiming to simplify the method. A scouting phase was conducted to select a suitable protein that mimics the aluminum-adjuvanted vaccine antigens’ UV/Vis optical signal. This approach simplifies the process and reduces costs, as commercially available proteins are more affordable than those in vaccine products, making the method appealing to quality control laboratories. These labs can purchase and qualify large quantities of commercial protein. Statistical evaluation tools, in accordance with the International Council for Harmonisation (ICH) guidelines, were used to define the critical method parameters during the screening phase. Screening designs investigated batch-to-batch differences in AH and sample concentrations, aligning with a quality by design (QbD) approach.

## 2. Materials and Methods

### 2.1. MenB Vaccine and Aluminium Hydroxide Formulation

The vaccine used to demonstrate the proof of concept of this method is the MenB vaccine (Table 1), a multicomponent vaccine for the prevention of serogroup B meningococcal disease (MenB) formulated with three recombinant antigens, 287-953 (NHBA), 961c (NadA), and 936-741 (fHbp), derived from serogroup B Neisseria meningitidis and an outer membrane vesicle (OMV) expressing porin A protein derived from New Zealand serogroup B *N. meningitidis* bacteria [5,6,7,11,12]. The three recombinant antigens are almost entirely adsorbed to AH, the latter being produced in-house by the GSK.

### 2.2. Microplate Readers

AH quantification was performed using the spectrophotometer Varioskan LUX Thermo Scientific that is a modular multi-technology plate reader and is controlled by Thermo Scientific SkanIt Software ver. 5.0.0.42 for Microplate Readers. The instrument also includes a plate shaking tool facilitated by an orbital shaking mode. The orbital shaking function is used for shaking the microplate to mix the samples in such a way to keep the AH in suspension during the analysis. For all experiments, orbital shaking was set at 1200 rpm for 1 min. As the AH is an opalescent suspension and because of the absence of a single clear absorption peak, the wavelength at which the optical signal of the selected sample shows the highest significative response was selected comparing the ratio between the aluminum suspension spectra and its blank within the visible range. Through the analysis of several spectra, 400 nm was selected as the optimal wavelength to perform a specific quantification of aluminum corresponding to the maximum ratio between the signal of the sample and the corresponding blank. Figure 1 reports an example of raw spectra for MenB formulation with and without AH (left) and the results after the analysis of the ratios between spectra (right).

### 2.3. Protein Model Selection

Before introducing the table containing the purchased proteins and their respective manufacturers and codes, it is essential to provide a brief overview of the selected commercial proteins and the rationale behind their selection. This explanation will be further detailed in Section 3.1, “Protein Model Selection and Results”. In the process of selecting the proteins for our study, careful consideration was given to the various factors influencing their suitability as model antigens. These considerations, including isoelectric points (IEPs), phosphorylation status, and commercial availability, were pivotal in ensuring the representativeness of our chosen models. This selection process, elucidated comprehensively in Section 3.1, aimed to mimic the characteristics of MenB antigens effectively. Table 2 below shows the model proteins used in this study.

### 2.4. High-Throughput—Automated Liquid Handling Platform

Owing to the labor-intensive nature of standard curve preparation, numerous efforts have been made to automate the procedure. A Hamilton MicroLab STAR liquid handling automation platform was used to achieve this goal. All liquid handling was automated using a Hamilton MicroLab Star Robotic workstation, which included the preparation of standards curve, controls (blank), and sample resuspension (by pipetting). Processing time was less than 11 s per sample or ∼9 min per 96-well plate (32 samples in duplicate or 64 in single), which is then immediately ready for the spectrophotometer reading. The Hamilton robot has been programmed to be able to handle different types of containers including pre-filled syringes (PFSs).The robot is programmed to prepare the calibration curve (each point of the calibration curve is formulated independently without serial dilutions) from the stock solutions listed in Table 3, which is placed inside the deck. Standard curve formulated samples and blanks are loaded in the plate (200 μL/well) by a liquid handling system, and at the end, the operator transfers the plate into the spectrophotometer, which shakes the samples at 1200 rpm for one minute and reads them at 400 nm.

### 2.5. Performance Evaluation

The performance evaluation was designed to verify the following parameters.

#### 2.5.1. Precision

Precision verifies the degree of agreement between a series of measurements obtained from the multiple sampling of the same homogeneous sample. Precision has been assessed through the verification of the intermediate precision (variability within the same lab) and repeatability (variability under the same operative conditions) parameters.

#### 2.5.2. Accuracy

Accuracy verifies the agreement between the value obtained with the analytical method and the true value. The accuracy has been verified considering the specification range of AH for MenB products: 2.4 mg/mL–3.6 mg/mL.

#### 2.5.3. Linearity

Linearity has been assessed by the linear fitting of the response in terms of aluminum concentration vs. theoretical concentrations.

#### 2.5.4. Robustness

As part of the robustness study, samples with different levels of AH as in Table 4 are formulated varying the amount of the drug substances (DSs) around the target (Table 1). In detail, the following formulations are prepared:-5 samples having the 5 selected different concentrations of aluminum and containing 30% less of the DSs target concentration;-5 samples having the 5 selected different concentrations of aluminum and containing 30% more of the DSs target concentration.

The method has been verified in samples formulated with a different content of DSs compared to the target of ±30% (Table 4). One replicate of each sample has been included in this evaluation.

### 2.6. Performance Evaluation Design

The performance evaluation plan consists of 12 analytical sessions performed by three operators who, using the Hamilton robot, prepared the final plates (4 plates for each operator where each analytical session corresponds to a prepared plate). In each session, two replicates of the following samples were tested. The samples are referred to as the formulation containing the five different concentrations of aluminum and DSs at the target concentration. The same lot of aluminum was used to build the STD curve and to formulate the formulations. The design of the experiments is schematized in Table 5. As indicated, two different ovalbumin lots are used in the experiments, and two different lots of 96-well plates were considered.

### 2.7. Bridging with an Orthogonal Method (The Titration Method)

As a further evaluation, a UV/Vis-based method was compared with the titration-based method, currently used for the release testing of aluminum content. Five different batches of MenB were produced at five different concentration levels of AH. From each batch, 6 samples were generated of which 3 were analyzed by the official release method for the quantification of AH at the quality control (QC) laboratories, and 3 samples were tested with the newly developed method in the R&D laboratory.

## 3. Results and Discussions

### 3.1. Protein Model Selection and Results

The aluminum at the surface of AH is coordinated with amphoteric hydroxyls that can accept or donate a proton depending on the pH of the solution [13]. This results in the AH adjuvant possessing a pH-dependent surface charge with a positive charge at neutral pH due to its isoelectric point (IEP) of 11.4. The adsorptive behavior of the AH adjuvant is greatly influenced by the composition of the buffers in which it is utilized because of aluminum’s high affinity for phosphate anions, moderate affinity for sulfate anions, and low affinity for other anions such as chloride and nitrate. The adsorption of proteins to solid surfaces, including aluminum adjuvants, is predominantly due to hydrophobic, electrostatic, and ligand exchange mechanisms. As a result, predicting the adsorptive behavior of antigens on aluminum adjuvants is challenging. Electrostatic interactions are one of the primary mechanisms of antigen adsorption to aluminum adjuvants, and they occur when the antigen and adjuvant have opposing charges. The positive surface charge of AH at neutral pH enables the electrostatic adsorption of antigens to it. Ligand exchange is another significant mechanism that drives the interaction between aluminum adjuvants and antigens; in fact, phosphate groups can exchange with hydroxyl groups at the surface of aluminum adjuvants, and antigens containing terminal phosphate groups can bind to aluminum adjuvants via ligand exchange. This mechanism can even overcome electrostatic repulsion. Based on these observations, the selection of model proteins to mimic MenB antigens took into account various features such as IEP or phosphorylation (Table 2).

All calibration curves obtained with the different model proteins were compared to the standard curve containing the MenB antigens. In all the calibrations reported here (Figure 2), the marked black line indicates the calibration curve performed with the MenB antigens (recombinant proteins and OMVs), and it can be seen that the angular coefficient and intercept remain approximately the same in all tests performed. Ovalbumin (b) and phosphorylated alpha-casein (h) show very good parallelism with the MenB formulation. These two proteins have a similar IEP and degree of phosphorylation. In fact, different results were obtained using dephosphorylated alpha-casein (f). Phosvitin (g) also shows a slope similar to that obtained with ovalbumin and alpha-casein. The other proteins tested did generate very different slope lines and, consequently, were not considered suitable candidates as model proteins to simulate MenB antigens. The result obtained for lysozyme was expected as it has an IEP very similar to that of AH. Although both alpha-casein and ovalbumin showed excellent performance, ovalbumin was chosen as the model protein to be used for the method set-up and qualification because of its ready availability and lower price. The equations for the lines depicted in these figures are provided in Table 6 (reference MenB antigens sample is underlined).

### 3.2. Performance Results

As reported in Section 2.5, AH content was determined in formulations designed to mimic MenB. These formulations used ovalbumin as a model protein in place of the MenB antigens, maintaining the same total protein concentration of 350 mcg/mL as the MenB vaccine. The results of each analytical run are reported in Appendix A (Table A1); information on the method’s performance was derived from these results.

Precision was tested in terms of repeatability (R) and intermediate precision (IP). The results showed that the maximum IP %CV was 4.1% and the maximum R %CV was 3.4% within the aluminum range considered.

Accuracy was tested for each aluminum level in terms of the relative bias % (RB%) from the theoretical level:RB% = Measured Concentration (mg/mL) − Theoretical Concentration (mg/mL)/Theoretical Concentration (mg/(mL) × 100

In brief, 90% confidence intervals for the mean of the RB% indicate that the accuracy, as trueness, was within ±5%, while 95% prediction intervals for the RB% indicate that accuracy, as total error, was within ±10%.

The linearity of the aluminum content response was verified through a linear fit of the measured concentration vs. theoretical concentration across all the analytical sessions. The R-squared was 0.986, and no pattern in the residuals was evident.

Finally, a robustness study demonstrated that the method was still suitable to determine aluminum content in samples formulated with a different content of DSs compared to the target (±30%).

Precision was evaluated at each concentration level (i.e., for each formulation sample). A mixed model was applied to the response variable (mg/mL of alum content) considering the operator and session (nested within the operator) as random factors. A variance decomposition was performed to estimate repeatability and intermediate precision; the results are presented in Table 7. The residual component indicates the variability in the replicate within each plate in each session. The residual is the component that determines repeatability, while the total indicates the intermediate precision. The CV% of the repeatability and intermediate precision associated with each level are reported in Table 7. The precision results show that both repeatability and intermediate precision were below 5%.

The RB was calculated using the same formula as reported above. Accuracy (or trueness) was evaluated by the 90% confidence interval of the mean RB% (as suggested by the guidelines), as reported in Table 8 and graphically represented in Figure 3. As is shown, all the results are within +/−5%.

**Figure 3 vaccines-13-00105-f003:**
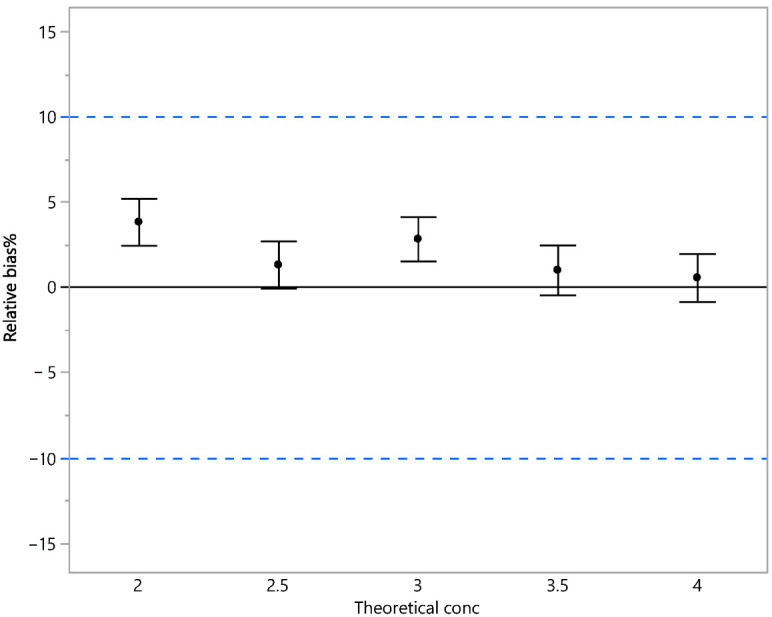
Relative bias % and 90% confidence interval (trueness). Black dots indicate the mean of the RB%; the vertical bars identify its 90%CI. The black horizontal line is 0% bias, and the blue horizontal dashed lines indicate ±10% bias. Accuracy was also evaluated in terms of total error using 95% prediction intervals of the RB% indicating the range in which an observation is expected to fall. The results obtained are within +/−10% as reported in Table 9 and graphically represented in Figure 4.

**Table 9 vaccines-13-00105-t009:** Accuracy evaluation using 95% prediction intervals.

Theoretical Conc.	Mean	Low PI95%	Up PI95%
2.0	3.85	−2.26	9.96
2.5	1.33	−4.81	7.47
3.0	2.85	−2.87	8.56
3.5	1.02	−5.45	7.49
4.0	0.57	−5.69	6.83

**Figure 4 vaccines-13-00105-f004:**
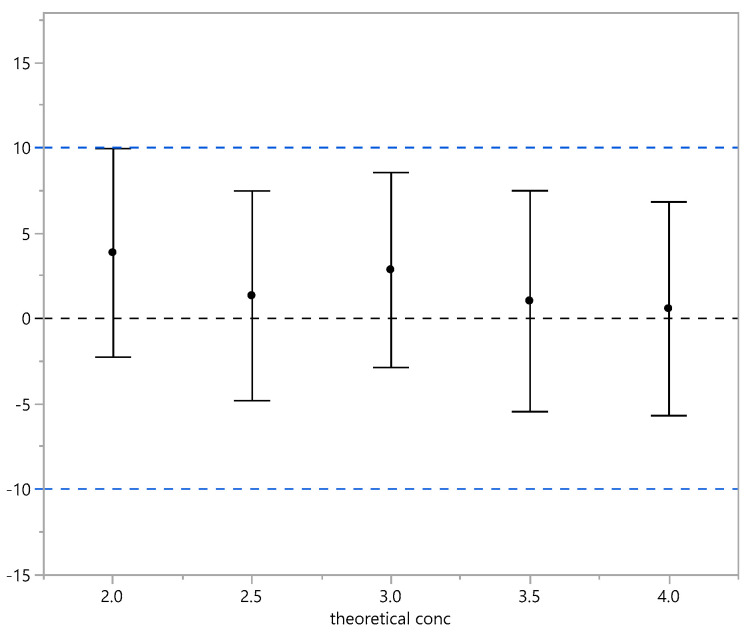
Relative bias % and 95% prediction interval (total error). Black dots indicate the mean of RB%; the vertical bars identify its 90%CI. The black horizontal line is 0% bias, and the blue horizontal dashed lines indicate ±10% bias.

In general, the linearity of an analytical procedure is its ability (within a given range) to obtain test results that are directly proportional to the concentration (amount) of analyte in the sample. For this study, linearity is demonstrated if the measured alum concentration increases proportionally as the theoretical concentration level increases. To this end, the mean of the two replicates of each level/session is considered, and the R-squared of the linear regression analysis and residual pattern was evaluated (Figure 5).

From the results in Figure 5, the R-squared of the linear fit is 0.986. In addition, no pattern in the residuals is visible from the residual plots, and no deviation from the normality assumption is evident from the normal quantile plot.

As described above, the method’s robustness has been verified in samples formulated with a different content (±30%) of DSs compared to the target (Table 4). The results of this experiment demonstrated that the method is capable of adequately determining the aluminum content in formulations where the amount of DSs varies due to normal process variability. The data of each run are reported in Appendix A (Table A2).

It should be noted that the samples considered in this study represent extreme cases in which all four DSs varied by the same percentage and in the same direction for a total of +/−30% (again 30% from the target is a very high variation that is unlikely to occur with the performance of the DS method). This is advantageous for the method because in the event of an error in the addition of an antigen in the formulation process, the method is still able to determine the aluminum content correctly.

Intermediate precision (Table 10) was calculated for each sample, and the highest CV was 5.4% in line with the variability observed in the precision study (Appendix A Table A1).

An accuracy evaluation (trueness) was also performed, and the results showed that the calculated RB% and 90% confidence intervals demonstrate that accuracy is within the ±10% considered acceptable (Table 11).

### 3.3. Bridging Results

The comparison between spectrophotometric method and the official release method showed comparable results (Figure 6), confirming the potential implementation of this new method as a possible release test for AH-adjuvanted products.

Overall, the performance outcomes of our spectrophotometric method suggest that it is a promising alternative for aluminum quantification in vaccine formulations, and it as an effective tool for both research laboratories and production environments. The positive correlation of results between our method and standard techniques such as EDTA back-titration and AAS further validates its potential as a reliable method in the ongoing monitoring and control of aluminum levels in adjuvanted vaccines. By using these proteins, we were able to create an analytical environment that closely mirrors the binding interactions between aluminum adjuvants and antigenic proteins. This approach has demonstrated a robust performance, underscoring its reliability and suitability for both research and quality control settings. The main limitations of this spectrophotometric method lie in the fact that a preliminary screening phase will always be necessary in order to find the correct model protein that mimics the antigens present in the vaccine to be tested. At present, this method has only been used for MenB-like formulations, and further studies will follow to assess the applicability to other vaccines with different adjuvant-uptake characteristics by finding the best model protein among those already evaluated while also considering other commercial proteins.

## 4. Conclusions

Aluminum salts, such as AH and aluminum phosphate, are the most commonly used adjuvants in vaccines due to their proven ability to enhance immune response. These compounds are utilized in vaccines because of their capacity to act as both a physical depot for antigens and as immunostimulants [14,15]. This dual role makes aluminum an essential component of many vaccines, especially those used to prevent diseases like hepatitis B, diphtheria, tetanus, and pertussis [16,17,18]. Regulatory bodies, such as the US Food and Drug Administration (FDA) and the European Medicines Agency (EMA), have established strict guidelines limiting the amount of aluminum in vaccines to ensure patient safety [19,20]. Monitoring aluminum content is necessary because exceeding these limits can lead to adverse effects, particularly in vulnerable populations such as infants, who are more sensitive to aluminum exposure. On the other hand, ensuring a sufficient aluminum content is equally important to guarantee the vaccine’s immunostimulatory function. The current release applied methods, such as ICP-OES and AAS, present several limitations such as being time-consuming, labor-intensive, and requiring specialized equipment and trained personnel; in addition, they cannot be applied in high-throughput mode, as is often required during vaccine development, where formulations must be tested, adjusted, and retested quickly to identify the optimal composition. The newly developed method, here described, was not only fast and efficient but also demonstrated resilience to the normal variations in process parameters, such as changes in antigen concentration. We demonstrated that the method is not sensitive to antigen concentration fluctuations of up to 30%, ensuring that the results remain consistent even under typical manufacturing variations. This robustness makes it a reliable tool for quality control throughout the vaccine production process. In conclusion, the proposed method offers a promising solution to the challenges currently faced in the quantitative analysis of aluminum in vaccines. Its superior speed, ease of use, and reduced environmental impact make it a valuable tool for both vaccine development and routine quality control in the pharmaceutical sector. Given the increasing demand for rapid, efficient, and environmentally friendly testing methods, this innovation represents a significant step forward in the optimization of vaccine production processes, helping to ensure that vaccines are both safe and effective while meeting the stringent requirements set by regulatory authorities.

## 5. Patents

Di Meola L. et al. Method to measure the concentration of AH in MenB vaccines. Publication Number WO/2023/067086 GLAXOSMITHKLINE BIO-LOGICALS SA [BE]/[BE].

## Figures and Tables

**Figure 1 vaccines-13-00105-f001:**
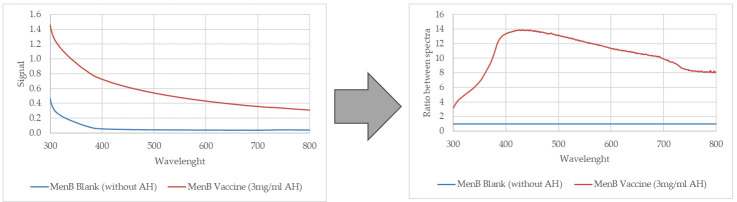
Included in the SkanIT software ver. 5.0.0.42, the tool «Ratio Between Spectra» allowed us to identify within the visible range the wavelength at which the optical signal of the selected formulation (MenB) showed the highest significative response compared to its blank. On the left is the spectrum of MenB adjuvanted with AH (in red) and the corresponding MenB without AH (in blue); on the right are shown the same spectra processed as the ratio between spectra. This processing highlights the maximum distance between the two at 400 nm.

**Figure 2 vaccines-13-00105-f002:**
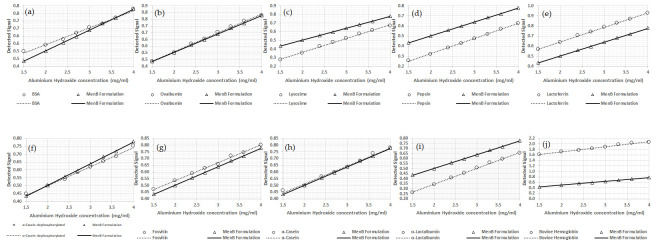
All calibrations performed with the different proteins where the abscissa axis shows the AH concentration, and the ordinate axis shows the signal obtained at 400 nm. In all figures, the marked black line indicates the calibration curve performed with the MenB antigens, while the proteins are presented in the following order: BSA (**a**); ovalbumin (**b**); lysozyme (**c**); pepsin (**d**); lactoferrin (**e**); dephosphorylated alpha-casein (**f**); fosvitin (**g**); alpha-casein (**h**); alpha-lactalbumin (**i**); bovine hemoglobin (**j**).

**Figure 5 vaccines-13-00105-f005:**
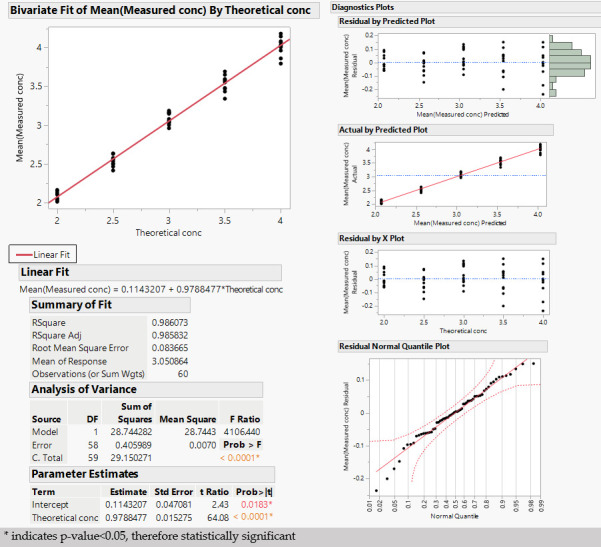
Linearity (colored dots identify different operators).

**Figure 6 vaccines-13-00105-f006:**
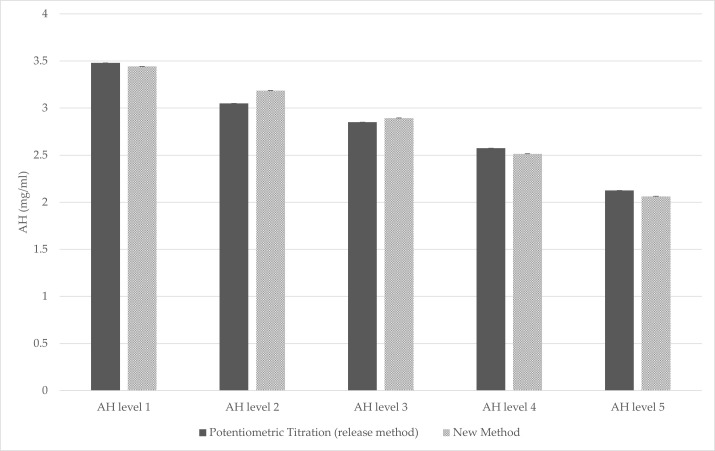
Graph showing the comparison between the data obtained with the MenB release test and the data obtained after the setup of the newly developed method.

**Table 1 vaccines-13-00105-t001:** MenB vaccine composition.

Component	Concentration
AH	3 mg/mL
NaCl	6.25 mg/mL
Histidine pH 6.3	10 mM
936-741	100 mcg/mL
961c	100 mcg/mL
287-953	100 mcg/mL
OMV	50 mcg/mL
Sucrose	2.0% *w*/*v*

The pH of the vaccines was maintained at 6.5 (±0.5), and the osmolarity was maintained at around 0.300 Osmol/Kg.

**Table 2 vaccines-13-00105-t002:** Candidate proteins.

Protein	Product Code	IEP	Molecular Weight (kDa)	Amino Acid Residues	Number of Potentially Phosphorylated Residues
Ovalbumin	Sigma Cod.A7641	4.5	42.7	385	Yes (1–2)
BSA	Sigma Cod.A9647	4.7	66.5	583	No
Lysozyme (from chicken egg)	Sigma Cod.L6876	10.5–11.0	14.6	129	No
Alpha-casein	Sigma Cod.C6780	4.2–4.8	22.0–25.0	195	Yes (1–8)
Dephosphorylated alpha-casein	Sigma Cod.C8032	4.6	25	195	No
Phosvitin	Sigma Cod.P1253	4	35	217	Yes (~80)
Alpha-lactalbumin	Sigma Cod.L6010	4.2–4.5	14.2	123	No
Bovine hemoglobin	Sigma Cod.H2500	6.8	64.5	574	No
Pepsin	Sigma Cod.P7000	3.24	34.5	326	No
Lactoferrin	Sigma Cod.L9507	8.7	80	700	No

**Table 3 vaccines-13-00105-t003:** Start solution. This table lists the two starting solutions to be placed inside the Hamilton robot with which it will independently formulate all calibration curve points and negative controls. With these two solutions in all formulations, only the AH content will be changed, leaving the content of all other components unchanged.

Solution Number	Solution Components	Component Concentration
1. Stock solution Std Curve preparation	AH	AH 5 mg/mL;
	NaCl;	NaCl 6.25 mg/mL;
	Histidine;	Histidine pH 6.5 10 mM;
	Sucrose;	Sucrose 2%
	Protein	Protein 350 mcg/mL
2. Diluent Solution Std Curve and control (Blank) preparation	NaCl;	NaCl 6.25 mg/mL;
	Histidine	Histidine pH 6.5 10 mM;
	Sucrose	Sucrose 2%
	Protein	Protein 350 mcg/mL

**Table 4 vaccines-13-00105-t004:** List of prepared formulations with different concentrations of aluminum and containing +/−30% of the DS target concentration. The formulations were manually prepared by the sequential addition of each component.

DS Concentration (∆Target)	Formulation Samples	Al(OH)3 Concentration (mg/mL)
−30%	1 LOW	2.0
−30%	2 LOW	2.5
−30%	3 LOW	3.0
−30%	4 LOW	3.5
−30%	5 LOW	4.0
+30%	1 HIGH	2.0
+30%	2 HIGH	2.5
+30%	3 HIGH	3.0
+30%	4 HIGH	3.5
+30%	5 HIGH	4.0

**Table 5 vaccines-13-00105-t005:** Scheme of the performance design with different concentrations of aluminum and DSs at the target concentration.

Operator	Ovalb.Lot	Plate Lot	Session	Replicate	Samples				
1, 2, or 3	1	1	1	1	1	2	3	4	5
				2	1	2	3	4	5
	2	2	2	1	1	2	3	4	5
				2	1	2	3	4	5
	1	1	3	1	1	2	3	4	5
				2	1	2	3	4	5
	2	2	4	1	1	2	3	4	5
				2	1	2	3	4	5

**Table 6 vaccines-13-00105-t006:** Equations obtained for each protein tested and comparison with the MenB antigens underlined.

Protein	R2	Equation
Ovalbumin	0.998	Y = 0.1406 + 0.2265
BSA	0.994	Y = 0.1126x + 0.3182
Lysozyme (from chicken egg)	0.997	Y = 0.1602x + 0.0400
Alpha-casein	0.991	Y = 0.1338x + 0.2446
Dephosphorylated alpha-casein	0.995	Y = 0.1211x + 0.2570
Fosvitin	0.996	Y = 0.1328x + 0.2705
Alpha-lactalbumin	0.999	Y = 0.1579x + 0.2940
Bovine hemoglobin	0.982	Y = 0.1893x + 1.3284
Pepsin	0.999	Y = 0.151x + 0.0221
Lactoferrin	0.999	Y = 0.1436x + 0.3557
MenB antigen reference sample	0.999	Y = 0.1374x + 0.2275

**Table 7 vaccines-13-00105-t007:** Precision evaluation performed at each of the 5 levels tested using variance decomposition to estimate repeatability and intermediate precision.

Theoretical Alum Concentration	Mean	Random Effect	Variance Component	SD	CV% Repeatability	CV% Intermediate Precision
2.0 mg/mL	2.1	operator	0.0024871	0.049870833		
		operator[session]	0	0		
		Residual	0.0048873	0.069909227	3.4%	
		Total	0.0073744	0.085874327		4.1%
2.5 mg/mL	2.5	operator	0.0016754	0.04093165		
		operator[session]	0.0028267	0.053166719		
		Residual	0.0008967	0.029944949	1.2%	
		Total	0.0053988	0.073476527		2.9%
3.0 mg/mL	3.1	operator	0.0006915	0.026296388		
		operator[session]	0.0042188	0.06495229		
		Residual	0.0017581	0.041929703	1.4%	
		Total	0.0066683	0.08165966		2.6%
3.5 mg/mL	3.5	operator	0.0009079	0.030131379		
		operator[session]	0.0076644	0.087546559		
		Residual	0.0028679	0.053552778	1.5%	
		Total	0.0114402	0.106958871		3.0%
4.0 mg/mL	4.0	operator	0	0		
		operator[session]	0.0079019	0.088892632		
		Residual	0.0080927	0.089959435	2.2%	
		Total	0.0159947	0.126470155		3.1%

**Table 8 vaccines-13-00105-t008:** Accuracy (trueness) evaluation.

Theoretical Level	Mean (Conc)	Mean (RB%)	Low CI90%	Up CI90%
2.0	2.077	3.85	2.47	5.23
2.5	2.533	1.33	−0.06	2.72
3.0	3.085	2.85	1.55	4.14
3.5	3.536	1.02	−0.44	2.48
4.0	4.023	0.57	−0.84	1.99

**Table 10 vaccines-13-00105-t010:** IP for samples tested in the robustness study.

Sample/Level	Theoretical Alum Conc. (mg/mL)	Mean	IP—CV%
1 LOW	2.0	1.92	5.4
2 LOW	2.5	2.43	4.4
3 LOW	3.0	2.98	3.8
4 LOW	3.5	3.52	3.2
5 LOW	4.0	3.99	4.0
1 HIGH	2.0	2.16	2.9
2 HIGH	2.5	2.64	3.0
3 HIGH	3.0	3.12	3.2
4 HIGH	3.5	3.46	2.7
5 HIGH	4.0	3.96	2.2

**Table 11 vaccines-13-00105-t011:** Accuracy evaluation (trueness).

Sample/Level	Theoretical Alum Conc. (mg/mL)	Mean	Mean (RB%)	Low CI90%	Up CI90%
1 LOW	2.0	1.92	−4.2	−6.88	−1.51
2 LOW	2.5	2.43	−3.0	−5.19	−0.79
3 LOW	3.0	2.98	−0.6	−2.52	1.42
4 LOW	3.5	3.52	0.7	−1.03	2.33
5 LOW	4.0	3.99	−0.3	−2.39	1.77
1 HIGH	2.0	2.16	8.0	6.37	9.56
2 HIGH	2.5	2.64	5.5	3.89	7.15
3 HIGH	3.0	3.12	4.0	2.34	5.75
4 HIGH	3.5	3.46	−1.1	−2.42	0.32
5 HIGH	4.0	3.96	−1.0	−2.10	0.17

## Data Availability

The data presented in this study are available on request from the corresponding author due to (specify the reason for the restriction).

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
