# Peer review of "New High-Throughput Method for Aluminum Content Determination in Vaccine Formulations"

_vaccines, 2025, doi:10.3390/vaccines13020105_

Round 1

Reviewer 1 Report

Comments and Suggestions for Authors

The authors presented a method for the quantitative determination of aluminum hydroxide in aluminum-adjuvanted vaccines. Unfortunately, the manuscript is difficult to understand and needs significant revision. The experiments with both model proteins and the MenB vaccine are not described clearly enough. The description of the statistical processing is also difficult to understand.

Specific comments:

Lanes 37-39 and 54-56 should be merged.

Lanes 77-102. This is a very long description of the advantages of the developed method, please shorten it. Briefly describe what your method is based on.

Figure 2. Please increase the font size for the figures. The legends are not readable.

The manuscript contains a large number of tables, which makes it difficult to understand the manuscript. Some of the tables contain raw data. Please move some tables to Supplementary Materials.

Figure 6. Please indicate standard deviation. Specify that 1-5 is given.

Author Response

Comments 1and 2: Lanes 37-39 and 54-56 should be merged. Lanes 77-102. This is a very long description of the advantages of the developed method, please shorten it. Briefly describe what your method is based on.

Responses 1 and 2: The text of the introduction was revised and summarised according to the reviewer's suggestion. In particular, lines 37-39 and 54-56 were merged while the text from line 77-102 was summarised (revising the entire paragraph).

Comment 3: Figure 2. Please increase the font size for the figures. The legends are not readable.

Response 3: The font size for figure 2 was increased and uploaded in the manuscript.

Comment 4: The manuscript contains a large number of tables, which makes it difficult to understand the manuscript. Some of the tables contain raw data. Please move some tables to Supplementary Materials.

Response 4: According to the reviewer’s suggestion, tables 7 and 11 containing raw data have been moved in the appendix A as a supplementary material.

Comment 5: Figure 6. Please indicate standard deviation. Specify that 1-5 is given.

Response 5:  Figure 6 has been reviewed and std deviation was added to the graph.

Reviewer 2 Report

Comments and Suggestions for Authors

The authors described a simple, innovative, non-destructive, high throughput method for the quantification of aluminum hydroxide in aluminum adjuvanted vaccines. The method was fully validated. However there are a few issues need to be clarified.

Comments:

1.     Please provide a representative spectrum for each AH standard and give a brief description about how the standard curve was calculated. 

2.     Please check the data in Table 7 and Table 11

Table 7 operator 1, Ovalb lot 2. Plate lot 2, Session 2, Rep 1 S2 2.5 mg/ml result.

Table 11 operator 2, Ovalb lot 2. Plate lot 2, Session 8, Rep 1 low and high S5 4.0 mg/ml result.

3.     Accuracy: The text content about between Line 281-286 is similar to that of Line 305-310, please check.   

4.     The linearity of the assay: Please provide the linearity of each standard curves including their Equations and R square.

Comments on the Quality of English Language

Minor:

1.      Line 10 aluminum hydroxide →aluminum hydroxide (AH)

2.      Line 14 aluminium Hydroxide (AH) AH

3.      Line 15, 20 aluminium hydroxide AH

4.      Line 27 please check the spell of Aluminium based adjuvants (alum), please check the use of alum in the text.

5.      Line 27 aluminium hydroxide (AH), AH should be used thereafter in the text.

6.      Line 31 Drug Product (DP) first time, thereafter DP should be used in the text.

7.      Line 191 Drug Substances (DSs), thereafter DS should be used

8.      Line 228 isoelectric point (iep) IEP

9.      Line 243 iep IEP

10.   Line 393 inductively coupled plasma optical emission spectrometry (ICP-OES)ICP-OES

11.   Line 394 atomic absorption 394 spectroscopy (AAS)AAS

Please check the following sentences:

1.      Line 34-36: “Aluminium adjuvants are used in vaccines such as hepatitis A hepatitis B diphtheria-tetanus-containing vaccines Hemophilus influenzae type b, and pneumococcal vaccines.”

2.      Line 271-274: “ According to what reported in Section 2.5. activities have been performed for the determination of Aluminium Hydroxide content in MenB -like formulations intended as formulation containing ovalbumin model protein instead of MenB antigens at the same concentration of the sum MenB vaccine components i.e. 350mcg/ml).”

3.      Line 365-366 “Its efficiency, accuracy, it as an effective tool for both research laboratories and production environments.”

Author Response

Comment 1: Please provide a representative spectrum for each AH standard and give a brief description about how the standard curve was calculated

Response 1: As shown in Figure 1, the maximum absorption signal for aluminum hydroxide in the MenB adjuvanted product corresponds to 400 nm. For the calibration curve, all standard points were read at a single wavelength (400 nm), and the entire spectrum was not acquired for each standard point.

Comment 2:  Please check the data in Table 7 and Table 11

Table 7 operator 1, Ovalb lot 2. Plate lot 2, Session 2, Rep 1 S2 2.5 mg/ml result.

Table 11 operator 2, Ovalb lot 2. Plate lot 2, Session 8, Rep 1 low and high S5 4.0 mg/ml result.

Response 2: The typos have been corrected, and tables 7 and 11 were moved in the appendix A.

Comment 3:  Accuracy: The text content about between Line 281-286 is similar to that of Line 305-310, please check.  

Response 3: The text has been updated according to the reviewer.

Comment 4: The linearity of the assay: Please provide the linearity of each standard curves including their Equations and R square.

Response 4: The table (named now A1 appendix A) has been updated with the requested data.

Minor:

  1. Line 10 aluminum hydroxide →aluminum hydroxide (AH)
  2. Line 14 aluminium Hydroxide (AH) →AH
  3. Line 15, 20 aluminium hydroxide →AH
  4. Line 27 please check the spell of Aluminiumbased adjuvants (alum), please check the use of alum in the text.
  5. Line 27 aluminium hydroxide (AH), AH should be used thereafter in the text.
  6. Line 31 Drug Product (DP)first time, thereafter DP should be used in the text.
  7. Line 191 Drug Substances (DSs), thereafter DS should be used
  8. Line 228 isoelectric point (iep) →IEP
  9. Line 243 iep →IEP
  10. Line 393 inductively coupled plasma optical emission spectrometry (ICP-OES)→ ICP-OES
  11. Line 394 atomic absorption 394 spectroscopy (AAS)→AAS

Response: All comments have been addressed and updated in the text.

Sentences:

Comment 1: Line 34-36: “Aluminium adjuvants are used in vaccines such as hepatitis A hepatitis B diphtheria-tetanus-containing vaccines Hemophilus influenzae type b, and pneumococcal vaccines.”

Response 1: Aluminum adjuvants are used in various vaccines, including those for hepatitis A, hepatitis B, diphtheria-tetanus-pertussis (DTaP), Haemophilus influenzae type b (Hib), and pneumococcal disease. Vaccine manufacturers therefore routinely perform quality control analyses to determine aluminum content, typically via potentiometric titration

Comment 2: Line 271-274: “ According to what reported in Section 2.5. activities have been performed for the determination of Aluminium Hydroxide content in MenB -like formulations intended as formulation containing ovalbumin model protein instead of MenB antigens at the same concentration of the sum MenB vaccine components i.e. 350mcg/ml).”

Response 2: As reported in Section 2.5, AH content was determined in formulations designed to mimic MenB. These formulations used ovalbumin as a model protein in place of the MenB antigens, maintaining the same total protein concentration of 350 mcg/ml as the MenB vaccine

Comment 3: Line 365-366 “Its efficiency, accuracy, it as an effective tool for both research laboratories and production environments.”

Response 3: Overall, the performance outcomes of our spectrophotometric method suggest that it is a promising alternative for aluminium quantification in vaccine formulations, and it as an effective tool for both research laboratories and production environments.

Round 2

Reviewer 1 Report

Comments and Suggestions for Authors

The new additions to the manuscript made a big difference. The quality of the paper had improved, and all my questions were addressed. No more comments.